# A superelastochromic crystal

Toshiki Mutai [1✉], Toshiyuki Sasaki [2], Shunichi Sakamoto[2], Isao Yoshikawa[1], Hirohiko Houjou [1] & Satoshi Takamizawa [2✉]

Chromism—color changes by external stimuli—has been intensively studied to develop smart materials because of easily detectability of the stimuli by eye or common spectroscopy as color changes. Luminescent chromism has particularly attracted research interest because of its high sensitivity. The color changes typically proceed in a one-way, two-state cycle, i.e. a stimulus-induced state will restore the initial state by another stimuli. Chromic systems showing instant, biphasic color switching and spontaneous reversibility will have wider practical applicability. Here we report luminescent chromism having such characteristics shown by mechanically controllable phase transitions in a luminescent organosuperelastic crystal. In mechanochromic luminescence, superelasticity—diffusion-less plastic deformation with spontaneous shape recoverability—enables real-time, reversible, and stepless control of the abundance ratio of biphasic color emissions via a single-crystal-to-single-crystal transformation by controlling a single stimulus, force stress. The unique chromic system, referred to as superelastochromism, holds potential for realizing informative molecule-based mechanical sensing.

[1] Department of Materials and Environmental Science, Institute of Industrial Science, University of Tokyo, 4-6-1 Komaba, Meguro-Ku, Tokyo 153-8505, Japan.
[2] Department of Materials System Science, Graduate School of Nanobioscience, Yokohama City University, 22-2 Seto, Kanazawa-Ku, Yokohama, Kanagawa 236-0027, Japan. ✉email: mutai@iis.u-tokyo.ac.jp; staka@yokohama-cu.ac.jp

Chromism[1] has been intensively studied with organic materials to exploit their colorability, transparency, designability, and other advantages in response to various stimuli such as photons, heat, electric charge, vapor, and mechanical stress. Fundamental research has a fascination with chromism because color changes based on wavelength shifts in absorption, emission, or reflection correlate with stimuli–responsive structural changes of materials on the subnanometer to micrometer scale. Chromism is also attractive from a practical viewpoint since the color changes are easily detectable by the naked eye or simple spectroscopy.

The process of color changes induced by pressing, shearing, cutting, and other mechanical forces is called mechanochromism[2–7], which is particularly important because the most fundamental stimuli in nature are commonplace mechanical forces that can be generated without any special apparatus. Mechanochromism based on elastic deformation has been demonstrated in amorphous gel materials[8–10] and a few molecular crystals under high pressure (3–10 GPa)[11] to show continuous changes in structural periodicity and color, which has led to quantitative mechanical sensing. Mechanically induced defects can also induce mechanochromism[4]. These systems exhibited spontaneously reversible chromism, but most chromic processes demand another stimulus for reverse chromism, i.e., reversion to the initial color.

As we previously reported about the polymorph-dependent luminescence of organic crystals, the luminescence color depends on the difference in molecular packing in crystals, and no chemical reactions are involved. Information about such systems could prove fruitful for elucidating the crystal structure–property relationship[12,13]. Mechanochromism based on a phase transition, e.g., polymorphic[14] or crystal-to-amorphous[2,3], is spontaneously irreversible but shows a biphasic color change at a high resolution, which is useful for memory storage and sensing tiny forces.

In this context, the focus of this work is organosuperelasticity, which we first discovered in 2014[15]. Elasticity is a common physical property in the spontaneous shape recoverability of materials. Recently, research on the manner of elastic deformation of organic crystals has been intensive[16–20] despite a general perception of their brittleness. In contrast, superelasticity or more specifically plastic deformation with spontaneous shape recoverability is a minor and unusual physical property, except in special kinds of metallic solids called superelastic alloys and shape-memory alloys[21,22], and research is still in its infancy especially in organic crystals[15,23–30]. In the elastic deformation, the density-gradient distribution of components causes strain to accumulate, whereas the superelastic deformation can accommodate strain through orientation changes of domains upon phase transition or twinning; thus, superelastic deformation has a potential ability to abruptly change physical properties. Since superelastic materials were developed specifically for metal alloys, their electronic physical properties limited coupling to conductivity and, recently, magnetism[31]. In organosuperelasticity, organic solids even in the single-crystal state show spontaneously reversible mechanically induced phase transitions, indicating the potential functionality of superelasticity with dielectric and optical properties due to the organic materials' characteristics. Very recently, organoferroelasticity—diffusion-less plastic deformation leaving spontaneous strain without spontaneous shape recoverability—in luminescent organometallic crystals was reported[32]. The luminescence is stable and unchanged during the organoferroelastic mechanical twinning, giving no mechanochromism.

Ideally, superelastic solids have optical functionality, and the occurrences and amounts of the phase transitions can be easily controlled in real time as material strain caused by mechanical stress, enabling biphasic color switching with excellent controllability in direction, region, timing, and velocity (Fig. 1). We find that single crystals of the luminescent organic compound 7-chloro-2-(2′-hydroxyphenyl)imidazo[1,2-a]pyridine (7Cl)[13] show superelasticity with luminescent color changes.

## Results

**Polymorphs and solid-state luminescence of 7Cl.** Solid-state luminescence from **7Cl** derivatives is based on an excited-state intramolecular proton transfer (ESIPT) (Fig. 2a)[33–36], resulting in a large Stokes shift and suppression of aggregation-caused quenching. Note that even small changes in solid-state molecular arrangements can cause a large shift of the emission spectra of **7Cl** crystals, due to the formation of an unusual environment-sensitive zwitterionic keto form in the excited state[37]. Most ESIPT-luminescent molecules are emitted from a nonionic keto form via enol-to-keto tautomerization when excited. Actually, two polymorphic **7Cl** crystals, YG and YO, showed yellow-green and orange fluorescence, respectively, under ultraviolet (UV) light (365 nm) (Fig. 2b, c).

**Mechanical deformation of a YG and YO crystal.** Superelastic behaviors were observed in YG crystals sheared under an optical microscope equipped with polarizing plates (Supplementary Movies 1 and 2). Under polarized white (PW) light, the mother ($\alpha_{YG}$) domain was converted into a daughter ($\beta$) domain in

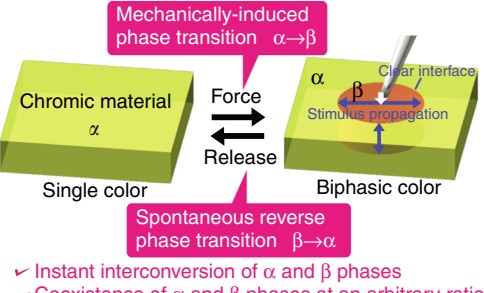

**Fig. 1 Concept of reversible and biphasic mechanochromism.** A single-color (absorption or emission) material in the mother ($\alpha$) phase shows biphasic colors superelastically when force is applied, indicated by a black arrow, along with a polymorphic phase transition from the $\alpha$ phase to a stress-induced daughter ($\beta$) phase.

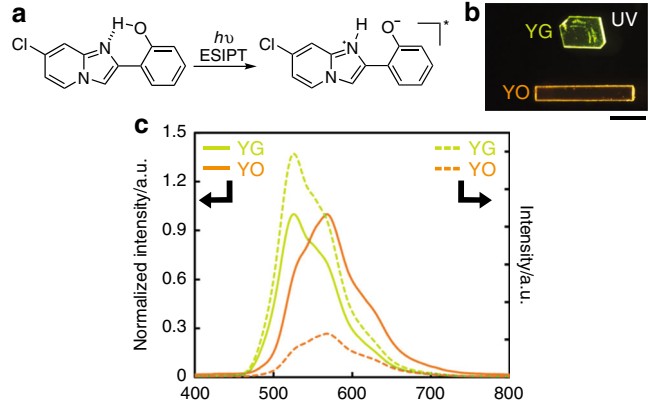

**Fig. 2 Fluorescent properties of 7Cl. a** Chemical structures of **7Cl** and schematic representation of ESIPT. **b** Photos of YG (top) and YO (bottom) crystals under UV light (365 nm) irradiation (scale bar: 400 μm). **c** Solid-state emission spectra of a YG crystal (yellow-green dotted line) and YO crystal (orange dotted line) excited by UV light (330–380 nm) at room temperature (r.t.). The solid lines show their normalized spectra.

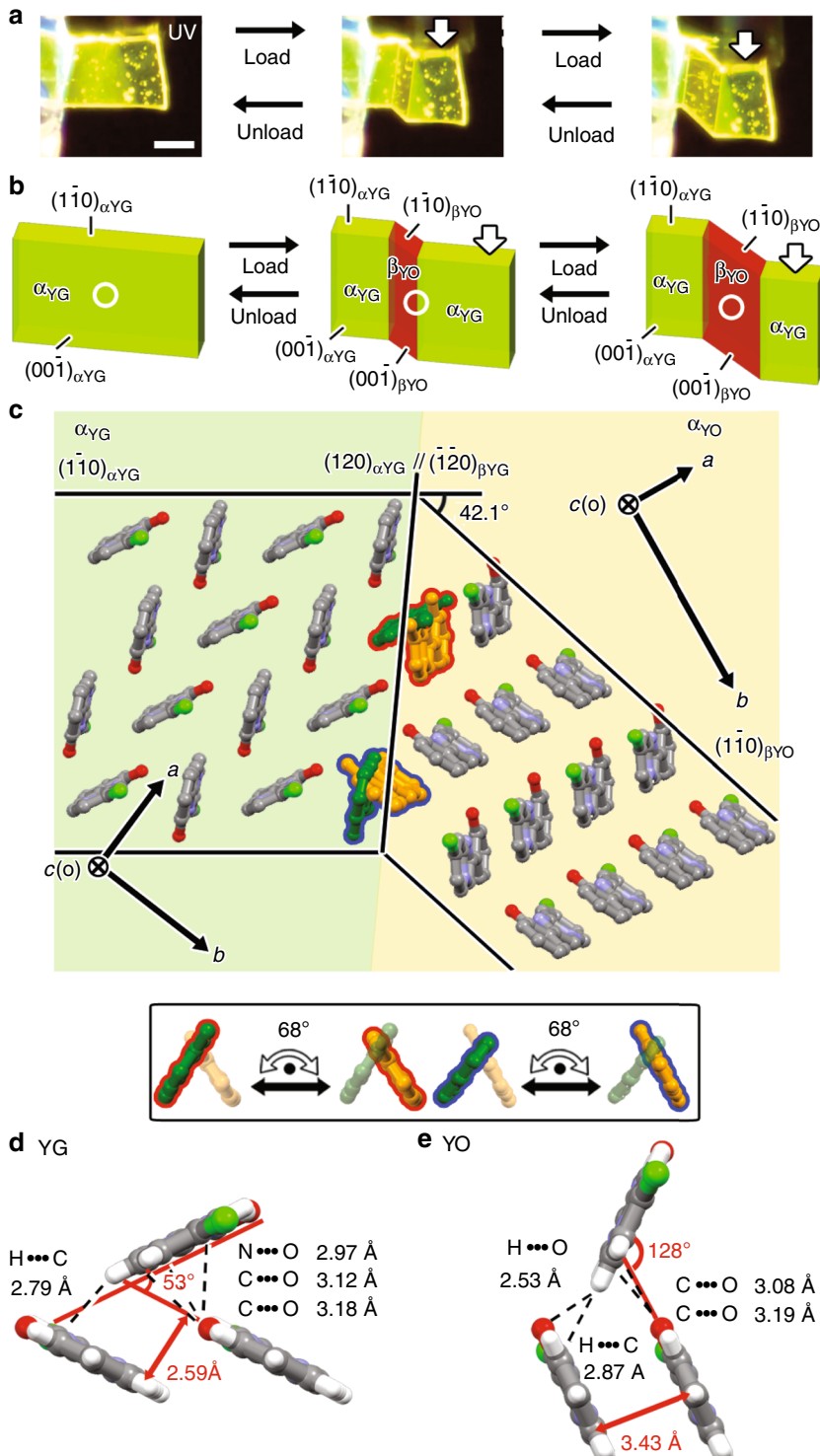

**Fig. 3 Superelasticity of a YG crystal. a** Snapshots of superelastic deformation of a YG crystal under UV light (365 nm) (scale bar: 100 μm). **b** Schematic representation of superelastic deformation and **c** estimated molecular correspondence at the $\alpha_{YG}//\beta_{YO}$ interface in a deformed YG crystal based on X-ray results. The white circles indicate areas of detected fluorescence in the in situ fluorescence spectroscopy in Supplementary Fig. 3. The inset represents possible molecular movements during the interconversion between the $\alpha_{YG}$ and $\beta_{YO}$ domains. The color of molecules at the interface is dark green ($\alpha_{YG}$) or orange ($\beta_{YO}$). The angles and distances are between neighboring **7Cl** molecules in the respective **d** YG and **e** YO polymorphs.

association with the deformation of a YG crystal (Supplementary Fig. 11). The deformed crystal spontaneously reverted to its initial shape when the β domain contracted after the shearing force was removed. Interestingly, the different fluorescence, yellow-green and orange, exhibited by the $\alpha_{YG}$ and β domains, respectively,

under UV light suggests that the β domain is the YO ($\beta_{YO}$) crystal (Fig. 3a). The conversion of the $\alpha_{YG}$ domain into the $\beta_{YO}$ domain in the superelasticity process was also confirmed by in situ fluorescence spectroscopy during the superelastic behavior (Supplementary Fig. 3) and by single-crystal X-ray diffraction

measurements of a YG crystal in the state with coexisting α and β domains (Supplementary Fig. 4, Supplementary Table 3). The superelasticity by conversion of the $\alpha_{YG}$ domain into the $\beta_{YO}$ domain was quite surprising because the transition between YG and YO polymorphs was a thermally non-inducible monotropic one according to differential scanning calorimetry (DSC) measurements (Supplementary Fig. 9). In general shape-memory alloys and some organosuperelastic materials, on the other hand, superelasticity is based on thermally reversible enantiotropic phase transition and shows a large temperature dependence (Supplementary Fig. 10). The mechanofluorochromism of YG crystals has spontaneous reversibility and controllability of fluorochromic regions originating from the characteristics of superelasticity in single crystals. Based on these features, the ability to sense mechanical force across a threshold was demonstrated by in situ fluorescence spectroscopy, during the superelasticity process, measuring changes in the ratio of two solid-state emissions correlating to the abundance ratio of the areas of the $\alpha_{YG}$ and $\beta_{YO}$ domains (Supplementary Fig. 3).

In the case of a YO crystal, a daughter ($\beta_{YG}$) domain confirmed by single-crystal X-ray diffraction measurements of a YO crystal in the state with coexisting α and β domains (Supplementary Fig. 5, Supplementary Table 4) was generated by shearing the mother ($\alpha_{YO}$) domain under both PW and UV light (Supplementary Fig. 11c, d). Here, a small $\beta_{YG}$ domain was also generated and spontaneously grew, suggesting that YG crystals are thermodynamically more stable than YO crystals at ambient conditions. Such clear and irreversible responses—crystal deformation and emission color change induced by a stimulus, a momentary mechanical force exceeding a threshold—are useful for sensing applications.

**Crystallographic studies of as-prepared and mechanically deformed crystals of 7Cl.** Molecular movement during the shear-induced phase transition between the two polymorphs was then investigated by X-ray crystallographic study of a superelastically deformed YG crystal (Fig. 3b–d). The interface of the two polymorphic domains was indexed as $(\bar{1}20)_{\alpha YG}//(120)_{\beta YO}$ (or $(120)_{\alpha YG}//(\bar{1}20)_{\beta YO}$), resulting in a calculated bending angle of 42.1°, which agrees well with the measurement by optical microscope observation (ca. 42°). The $\alpha_{YG}$ domain transformed into the $\beta_{YO}$ domain by a 68° (or 61° and 16°) rotation of 7Cl molecules, which also induced their displacive motions of 2.0 and 1.9 Å along the $a$- and $b$-axes, respectively, to optimize the herringbone arrangement at the interface (Fig. 3c–e, Supplementary Figs. S6–S8). The relatively large molecular rotation in comparison with previous examples of organosuperelastic materials—e.g., terephthalamide that rotates ca. 10°–32° during thermally or mechanically induced phase transition—is one possible reason for the monotropic nature of a 7Cl crystal. While anisotropic shear stress can effectively trigger the specific molecular movement required for the crystal transition, thermal activation of molecular rotation may not be practically sufficient to induce phase transition in this case. The emission color difference between the YG and YO crystals can be attributed to a difference in the arrangement of 7Cl molecules; for example, the π overlap of 7Cl molecules is larger in a YO crystal than a YG crystal, leading to red-shifted emission spectrum (Fig. 3d, e, Supplementary Fig. 6).

**Mechanical characterization of superelasticity in a YG crystal.** Stress-displacement curves recorded in shear tests revealed the effects of temperature and light on the mechanical properties of YG crystals (Fig. 4, Supplementary Figs. 12–17, Supplementary Table 6). The YG crystals deformed in the same way under all

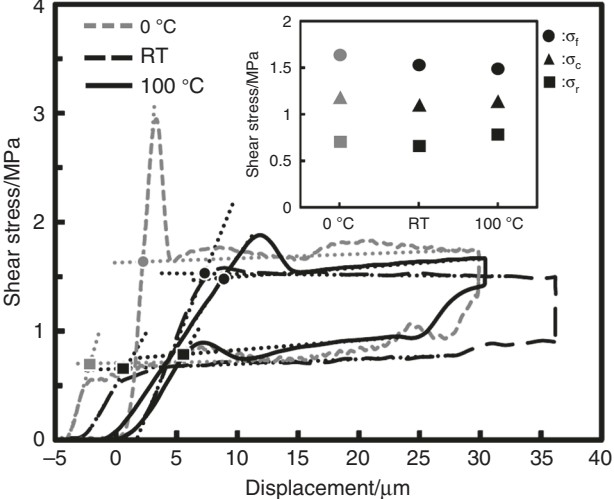

**Fig. 4 Shear tests on YG crystals.** Stress–displacement curves of YG crystals recorded under PW light at various temperatures: 0 °C (short-dashed line), r.t. (RT, medium-dashed line), and 100 °C (long-dashed line). The inset represents effective shear stresses ($\sigma_f$: circle, $\sigma_c$: triangle, and $\sigma_r$: square) under the conditions estimated from the intersections of dotted lines indicating elastic and plastic deformations, except for $\sigma_c$ calculated as a mean value of $\sigma_f$ and $\sigma_r$.

conditions: under PW light at 0 and 100 °C, in the dark, and under PW light and UV light irradiation at r.t.

Under PW light at r.t., a YG crystal showed a typical superelastic hysteresis loop; effective shear stress of 1.527 and 0.657 MPa for forward ($\sigma_f$) and reverse ($\sigma_r$) deformation, respectively; chemical shear ($\sigma_c$) of 1.092 MPa; energy-storage density ($E_s$) of 553.6 kJ m$^{-3}$; dissipated energy density ($E_d$) of 733.0 kJ m$^{-3}$; energy-storage efficiency ($\eta$) of 0.430; a superelastic index ($\chi$) of 0.505 (all these values were calculated from the experimental values in the stress-displacement curves) (Fig. 4 long-dashed line, Supplementary Table 3). The $\sigma_f$ and $E_S$ values are ca. 3 and 9 times, respectively, those of phase transition-based organosuperelasticity in a terephthalamide crystal[15] are ca. 22 and 47 times, respectively, which are those of twinning-based organosuperelasticity in a 3,5-difluorobenzoic acid crystal[23] (Supplementary Table 7). More importantly, the $\sigma_c$ values of 1.173 MPa at 0 °C (Fig. 4 short- dashed line, Supplementary Fig. 12) and 1.136 MPa at 100 °C (Fig. 4 solid line, Supplementary Fig. 14) are close to those measured at r.t., demonstrating temperature independence of the superelasticity attributable to a mechanically induced phase transition between monotropic polymorphs, which is different from that between enantiotropic polymophs in the superelasticity of shape-memory alloys (Fig. 4 inset, Supplementary Fig. 17). In addition, the mechanical parameters under UV light almost correspond to those measured under PW light, showing UV light independence of the superelasticity (Supplementary Fig. 16).

**Discussion**

In conclusion, ESIPT-fluorescent crystals of 7Cl, a pure and simple organic molecule, exhibit spontaneously reversible chromism due to a temperature-independent superelasticity, showing biphasic luminescent color switching with an emission intensity dependent on the domain volume and the sensing of small stress; for example, a YG crystal with an intersectional area of ca. 19 μm$^2$ can detect an *Aphaenogaster famelica* (ca. 3 mg). Organosuperelasticity offers the advantage of reversible and strict crystal deformation in association with molecular rearrangement against

applied force. Mechanochromism can be induced at any position (s) at any time by superelasticity, or mechanically well-regulated structural transition. These characteristics differentiate super-elastochromism from conventional chromisms, e.g., super-elastochromism can detect in situ mechanical stress quantitatively: detection of the moment when mechanical stress is applied and removed, and how much work is done on a crystal specimen. The advantage of ESIPT molecules is their strong emissions with a color sensitive to solid-state molecular arrangement. Combining these advantages will open up the science of superelastochromism, which can expand the designability and applicability of chromic materials in the future.

## Methods

**Synthesis of 7-chloro-2-(2′-hydroxyphenyl)imidazo[1,2-a]pyridine (7Cl)**. An acetonitrile (40 mL) solution of 2-(bromoacetyl)anisol (2.03 g, 8.86 mmol), 4-chloro-2-amino-pyridine (1.14 g, 8.86 mmol), and $NaHCO_3$ (1.55 g, 18.45 mmol) was refluxed for 20 h. After filtering off the insoluble solid, the filtrate was evaporated, and the residue was applied to a silica gel column ($CHCl_3$/ethyl acetate = 100:0 to 40:1) to afford 7-chloro-2-(2′-methoxyphenyl)imidazo[1,2-a]pyridine (1.69 g, 74%, m.p. 138–139 °C). Then a cooled solution (0 °C) of well-dried 7-chloro-2-(2′-methoxyphenyl)imidazo[1,2-a]pyridine (1.51 g, 5.84 mmol) in anhydrous dichloromethane was dropwise added to a dichloromethane solution (1.0 mol/L) of boron tribromide (18 mL, 3.1 eq.). The reaction mixture was allowed to reach room temperature and further stirred for 3 h. A saturated aqueous $NaHCO_3$ was slowly added with stirring, and then separated with water and chloroform. The organic layer was washed with water and brine, and then dried over $Na_2SO_4$. The organic layer was evaporated to give the crude product. Purification by recrystallization from ethanol (0.80 g, 56%). M. p.: 196–197 °C. [1]H NMR ($CDCl_3$, 400 MHz) $\delta$: 12.38 (1H, s, OH), 8.05 (1H, dd, $J = 4.5, 0.5$ Hz, 5-H), 7.81 (1H, s, 3-H), 7.59 (1H, sd, $J = 1.3$ Hz, 8-H), 7.54 (1H, dd, $J = 4.7, 1.0$ Hz, 6′-H), 7.23 (1H, ddd, $J = 5.0, 4.5, 1.0$ Hz, 4′-H), 7.02 (1H, dd, $J = 5.0, 1.0$ Hz, 3′-H), 6.87 (1H, ddd, $J = 5.0, 4.5, 1.0$ Hz, 5′-H), and 6.84 (1H, dd, $J = 4.3, 1.0$ Hz, 6-H) (Supplementary Fig. 18). [13]C-NMR ($CDCl_3$, 100 MHz) $\delta$: 157.3, 146.3, 143.4, 131.7, 130.1, 125.9, 125.6, 119.2, 117.9, 115.8, 115.8, 114.9, and 106.9 (Supplementary Fig. 19). Anal. Calcd for $C_{13}H_9ClN_2O$: C,65.00; H,4.29; N,10.83%. Found: C,64.86; H,4.26; N,10.89%.

**Crystal preparation**. Single crystals of **7Cl** were prepared by recrystallization from ethanol solutions, affording two polymorphic crystals: YG and YO. Single crystals in the YG form were selectively obtained from a **7Cl** solution in 1:1 mixture of THF and toluene at r.t.

**Microscope observations**. An optical microscope (SZ61, Olympus Co.) equipped with polarizing plates and a digital camera was used to record mechanical deformation of crystals using tweezers.

**DSC measurements**. DSC measurements of **7Cl** crystals were carried out using a DSC-60 (Shimadzu Co.) and DSC 7020 (Hitachi High-Technologies Co.) instruments under a nitrogen gas flow (65 ml $min^{-1}$). Crystals in YG and YO form with the size of a few hundreds of micrometers for the measurements were prepared by slow evaporation of a **7Cl** solution in 1:1 mixture of THF and toluene and in ethanol, respectively, at r.t. Experimental conditions are summarized in Supplementary Table 1.

**Single-crystal X-ray structural analysis**. A mechanically deformed YG (coexisting state of $\alpha_{YG}$ and $\beta_{YO}$ domains) and YO crystals (coexisting state of $\alpha_{YO}$ and $\beta_{YG}$ domains) were prepared in addition to as-prepared **7Cl** single crystals in YG and YO forms. To avoid spontaneous dissipation of a $\beta_{YO}$ domain, the mechanically deformed YG crystal was partially cleaved at the $\alpha_{YG}//\beta_{YO}$ interfaces. Single-crystal X-ray diffraction measurements of the crystals were performed at 298 K (25 °C) with a CMOS detector (Bruker Photon III C14) with a nitrogen-flow temperature controller using a rotating anode X-ray source (MoK$\alpha$ radiation ($\lambda$ = 0.71073 Å)). Multi-scan absorption corrections were applied using the SADABS program. The structure was solved by intrinsic phasing methods (SHELXT-2014/5) and refined by full-matrix least-squares calculations on $F^2$ (SHELXL-2016/6). Non-hydrogen atoms were refined anisotropically; hydrogen atoms were fixed at calculated positions by riding model approximation. With respect to the mechanically deformed crystals, X-ray diffraction patterns were obtained around the interface of $\alpha_{YG}//\beta_{YO}$ and $\alpha_{YO}//\beta_{YG}$ and they were analyzed as twins. Crystal face indexing was carried out using APEX III Ver.2016.1-0 program package with a twin-resolution program (Supplementary Figs. 4 and 5).

**Force measurements**. Shear tests were carried out on a universal testing machine. A crystal fixed on a glass base was sheared by a glass jig and observed under an optical microscope equipped with polarizing plates. Light sources: LA-HDF5010 (90W, HAYASHI-REPIC CO., LTD.) for VIS light and HLV-24UV365-4WPCLTL (365 nm, 0.7A/3.3W, CCS Inc.) connected to PD3-5024-4-EI(A) (CCS Inc.) for UV light. The experimental information and schematic representation of the setup are shown in Supplementary Table 2 and Supplementary Fig. 1, respectively.

## Data availability

All the data generated or analyzed during this study are included in this published article (and its Supplementary information files) or available from the authors upon reasonable request. The X-ray crystallographic coordinates for structures reported in this study have been deposited at the Cambridge Crystallographic Data Centre (CCDC), under deposition numbers 1969297–1969298. These data can be obtained free of charge from The Cambridge Crystallographic Data Centre via www.ccdc.cam.ac.uk/data_request/cif.

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

## Acknowledgements

Funding was provided by JSPS KAKENHI Grant numbers JP17H06367 (Grant-in-Aid for Scientific Research on Innovative Areas), JP16K5743 and JP19K05434 (Grants-in-Aid for Scientific Research (C)) for T.M., and the 2016–2018 Strategic Research Promotion (SK2810) fund of Yokohama City University and JSPS KAKENHI Grant numbers JP17H06368 (Grant-in-Aid for Scientific Research on Innovative Areas) and JP17K19143 (Grant-in-Aid for Challenging Research (Pioneering)) for S.T.

## Author contributions

T.M. and S.T. designed the project, examined the data, and edited the paper. T.M. synthesized the chemicals. T.S. analyzed the experimental data and prepared the paper. S.S. carried out the experiments on mechanical properties. I.Y. and H.H. carried out the experiments on luminescence.

## Competing interests

The authors declare no competing interests.
