## [Peer Review File · Nature Communications]

Reviewers' comments:

Reviewer #1 (Remarks to the Author):

This work is significant as it combines superelastic single-crystal-to-single-crystal transition with mechanochromic behavior which has not been reported before. The molecular system and its polymorph-based luminescent properties have been previously studied (esp. ref [13]). Mechanically induced polymorph transition is the new aspect of this work. One unique and yet puzzling aspect of this manuscript is that superelasticity is based on monotropic polymorphs, an observation that is poorly supported. This observation is in stark contrast to the thermo-mechanical behaviors of shape memory alloys and molecular crystals that transform in between enantiotropic phases. Overall, this referee believes that the authors' observations are rather unique and potentially suitable for publication in Nature Communications. However, there are several glaring technical issues that have to be addressed before it can be considered further, which are detailed below.

1) One major issue is that the important observation of superelastic transition based on monotropic systems is inferred from a single set of DSC data that is poorly analyzed and presented, and the details of the experiments are not even provided, such as how are the powders prepared, and what are the scanning rates etc. The melting peak of the YO crystal in Figure S8b is asymmetric and has a large tail at the onset of "melting". It is possible that the polymorph transition to YG is hidden in this peak, which can potentially be revealed if using single crystal samples and using very low scan rate. Further, at lower temperatures, the DSC scanning curve is not presented at high resolution. It is well known that polymorph transitions in organic crystals can give very small enthalpic change, that it can be missed in a poorly analyzed DSC curve. Therefore, alternative methods for observing thermally induced polymorph transitions must be carried out to validate this important claim.

Further, in order to check the single-crystal-single-crystal transformability of YG (YO) crystals, it would be better to run heating-cooling cycle between -140°C and ca. 150°C before melting. In particular, since the authors carried out thermo-mechanical tests at 0 – 100 °C, using YG and YO crystals, it is necessary to show no MS temperature appears below 0 °C under cooling process.

In addition, DSC curves show small endothermic transition peaks at ca. 0 - 20 °C for both polymorphs. What is the origin for these transition peaks?

2) Another glaring issue is the lack of mechanistic understanding behind this unique observation: how is it thermodynamically possible that superelastic transition can occur between two monotropic polymorphs?

3) Further, the authors stress once and again the potential application of this phenomenon in mechanical sensing. Yet the change in emission properties is rather small and hard to discern under microscope. Is this limitation intrinsic to superelastic transitions? Meaning that only small structural and thus property changes can afford single-crystal-to-single-crystal superelastic transitions? This aspect needs to be discussed in the manuscript.

4) Critically, all the experimental sections are way too brief missing important details that prevent other researchers in the field from reproducing their results. The manuscript is not suitable for publication until the methods are described with sufficient detail.

5) The authors claim exceptionally high effective shear stress and energy storage density compared with previous reported superelastic crystals. These data are based on a single measurement on one single crystal. Statistical analysis of these data as well as all other metrics in the same paragraph must be presented.

6) Since the superelasticity in monotropic system is new, it would be better to show schematic Gibbs free energy vs. variants (e.g., T) diagram in comparison to the enantiotropic system.

7) The authors provided crystal structure of before and after phase transition to show rearrangement of molecules in these structures. As the authors described, molecular packing change attributes emission property alteration, which is indicative of displacive motion of molecules in addition to 68° rotation of molecules upon transformation. The displacive term is not clearly stated in the manuscript.

Reviewer #2 (Remarks to the Author):

In this paper authors report two polymorphic forms of ESIPT active 7-chloro-2-(2'-hydroxyphenyl)imidazo[1,2-a]pyridine and elastic behavior of one polymorphic form. Authors claim that YG form get converts to YO form and vice versa. While YG form shows elastic nature, the other form does not. Authors claim that upon elastic stretching the YG form transforms to YO form (which I am not convinced with). Also, Mechanochromism in crystals via elastic deformation has already been known in the literature (authors cited some of the examples).

Different emission of different polymorphs of ESIPT active 2-Hydroxy imidazopyridinyl molecules are already reported by these authors (Angew. Chem. Int. Ed. 2008, 47, 9522). This polymorph dependent luminescence switching was done by heat as the stimuli. In the current paper mechanical force has been used as the trigger for polymorphic interconversion, which is a very well-known trigger for polymorphic transitions. This paper does not have the quality and scientific vigor required for a Nature communication article.

The elasticity in crystals is well known. I am surprised to see authors have not cited the work of Desiraju group and Malla Reddy group who have published several papers on elastic crystals.

The abbreviation for room temperature is r.t. not RT (RT is the Product of Universal gas constant and absolute temperature)

Authors state that "Interestingly, the different fluorescence, yellow green and orange, exhibited by the α YG and β domains, respectively, under UV light suggests that the β domain is the YO (β YO) crystal (Fig. 3a, Mov. S2)". But, as per the video S2 and Fig. 3a, beta domain does not exhibit orange fluorescence (more convincing evidences may be required).

How authors have determined the crystal structure of beta YG form? How authors got the elastically stretched form for crystal structure determination is not clear. The details of the data collection will help other researchers to reproduce this results.

Also, even if it is showing orange fluorescence, how can authors say it is the YO crystal?

The figure 3 is just based on speculation. Are there any evidences ?

Readability: this paper has poor readability which talks at length about the calculated data.

Experimental data are minimum.

Reviewer #3 (Remarks to the Author):

It is an exciting article by Takamizawa et al, which describes the superelastochromic nature in a biphasic organic crystalline material. The authors elegantly present the mechanical force induced phase transformation of a luminescent YG (yellow-greenish) polymorph to the YO (orange) polymorph in a single-crystal-to-single-crystal fashion, allowing the clear detection of the colour change. The structures of the two phases (using SCXRD technique) and the superelastic behaviour of the samples are very well characterized. The origin of superelasticity is nicely correlated to the underlying crystal structure of the two forms. The crystals are so sensitive to the mechanical force that even movement of an ant can be detected by the fluorescence sensing, which I feel would find

attractive practical applications. Hence, I feel the quality of the article is suitable for publication in Nature Communications.

I do not find any draw backs in the article, but feel that more citations related to mechanochromic luminescence (ML) needs to be added from other groups. For instance, the authors discussed about the ML from systems with phase transformations, but did not mention about those involving stress-induced defects (no-phase transformations) such as those involving avobenzene based brittle and plastic polymorphs, etc. Although, the article already covers many pertinent references for the context of the current work, the broader perspective covering other mechanically responsive crystals would enhance the interest.

I find the article free of errors and easy to follow.

Responses to reviewer comments for “A superelastochromic crystal”

Reviewer #1 (Remarks to the Author):

Thank you for your kind and valuable comments.

This work is significant as it combines superelastic single-crystal-to-single-crystal transition with mechanochromic behavior which has not been reported before. The molecular system and its polymorph-based luminescent properties have been previously studied (esp. ref [13]). Mechanically induced polymorph transition is the new aspect of this work. One unique and yet puzzling aspect of this manuscript is that superelasticity is based on monotropic polymorphs, an observation that is poorly supported. This observation is in stark contrast to the thermo-mechanical behaviors of shape memory alloys and molecular crystals that transform in between enantiotropic phases. Overall, this referee believes that the authors' observations are rather unique and potentially suitable for publication in Nature Communications. However, there are several glaring technical issues that have to be addressed before it can be considered further, which are detailed below.

Q.1-1. 1) One major issue is that the important observation of superelastic transition based on monotropic systems is inferred from a single set of DSC data that is poorly analyzed and presented, and the details of the experiments are not even provided, such as how are the powders prepared, and what are the scanning rates etc. The melting peak of the YO crystal in Figure S8b is asymmetric and has a large tail at the onset of “melting”. It is possible that the polymorph transition to YG is hidden in this peak, which can potentially be revealed if using single crystal samples and using very low scan rate. Further, at lower temperatures, the DSC scanning curve is not presented at high resolution. It is well known that polymorph transitions in organic crystals can give very small enthalpic change, that it can be missed in a poorly analyzed DSC curve. Therefore, alternative methods for observing thermally induced polymorph transitions must be carried out to validate this important claim.

Further, in order to check the single-crystal-single-crystal transformability of YG (YO) crystals, it would be better to run heating-cooling cycle between -140°C and ca. 150°C before melting. In particular, since the authors carried out thermo-mechanical tests at 0 – 100 °C, using YG and YO crystals, it is necessary to show no MS temperature appears below 0 °C under cooling process.

In addition, DSC curves show small endothermic transition peaks at ca. 0 - 20 °C for both polymorphs. What is the origin for these transition peaks?

A.1-1. After making further DSC measurements, we concluded that YG and YO

crystals are not thermally interconvertible with each other (under atmospheric pressure).

To assuage your doubts, we carefully conducted DSC measurements of YG and YO crystals again according to your concerns. In our additional observations, we not only could not find any signs of thermal transition in the temperature range from -140°C to the melting point but also confirmed no peak overlaps during melting. The revised DSC charts (Fig. S9) suggest that the peak around 0–10°C in the previous DSC charts is attributable to water from air, and the revised charts show that no polymorph transitions were missed.

Experimental details about the DSC measurements have been updated with Table S1 and Fig. S9 in the supplementary information: the degree of tailing in endothermic peaks during melting was smaller without the appearance of any additional peaks when the scanning rate was slowed to 0.5°C/min.

Table S1. DSC measurement conditions.

Crystal	Weight / mg	Scanning rate / °C min ⁻¹	Temperature range
YG ^a	2.78	6	30°C–220°C
YG ^b	4.79	5	-140°C–150°C
YG ^b	4.79	0.5	180°C–215°C
YO ^a	3.00	6	30°C–220°C
YO ^b	2.38	5	-140°C–150°C
YO ^b	2.38	0.5	180°C–215°C

Measurements were taken on the ^a DSC-60 (Shimadzu Co.) and ^b DSC 7020 (Hitachi High-Technologies Co.) instruments under a nitrogen gas flow (65 ml min⁻¹).

Figure S9. Differential scanning calorimetry measurements. Charts (a)–(c) show a YG crystal, and charts (d)–(f) show a YO crystal. The melting points of the YG and YO crystals were estimated at 197.7°C and 197.4°C, respectively.

Q.1-2. 2) Another glaring issue is the lack of mechanistic understanding behind this unique observation: how is it thermodynamically possible that superelastic transition can occur between two monotropic polymorphs?

A.1-2 We posited that shear stress can induce phase transition if a structurally metastable state exists. Fortunately, we were able to confirm two different crystal phases in reality through a mechanical evaluation of their relative differences in thermal stability, and a structural hindrance “practically” inhibits a smooth thermal transition (under atmospheric pressure). Therefore, the stress concentration by applying shear force on the crystal surface can work as an effective trigger for structural transition, and a uniform shear force can maintain the propagation of transition after phase induction. In this experiment, the monotropic (or thermally non-inducible) nature and the drastic color change along with the superelastic crystal deformation are most likely attributable to the large molecular movements, which as a structural hindrance inhibited the phase transitions. Although the structural difference between YG and YO forms seems to be small, the required rotation angle of 68° upon phase transition is considerably large. Furthermore, the changes in the herringbone arrangement and the lengths of the *a* and *b* axes in YG and YO crystals indicate slight displacive motions of 7Cl molecules. Note that the molecular movements can change the π -overlapping manner between neighboring 7Cl molecules in YG and YO crystals, allowing the color

switch.

We have added an explanation on these thoughts to the manuscript and modified Figure S6 in the supplementary information as follows:

“The α_{YG} domain transformed into the β_{YO} domain by a 68° (or 61° and 16°) rotation of **7CI** molecules, which also induced their displacive motions of 2.0 \AA and 1.9 \AA along the a and b axes, respectively, to optimize the herringbone arrangement at the interface (Figs. 3c–e, S6, S7, S8). The relatively large molecular rotation in comparison with previous examples of organosuperelastic materials—e.g., terephthalamide which rotates ca. 10° – 32° during thermally- or mechanically-induced phase transition—is one possible reason for the monotropic nature of a **7CI** crystal. While anisotropic shear stress can effectively trigger the specific molecular movement required for the crystal transition, thermal activation of molecular rotation may not be practically sufficient to induce phase transition in this case.”

Figure S6. Thermal ellipsoids (50% probability level), packing diagrams, and overlaps of **7CI** molecules in a (a) YG crystal and (b) YO crystal.

Q.1-3 3) Further, the authors stress once and again the potential application of this phenomenon in mechanical sensing. Yet the change in emission properties is rather small and hard to discern under microscope. Is this limitation intrinsic to superelastic transitions? Meaning that only small structural and thus property changes can afford

single-crystal-to-single-crystal superelastic transitions? This aspect needs to be discussed in the manuscript.

A.1-3. Our latest video (new updated Mov. S1) improves the appearance of the color change to the eye. We believe you can discern the colors of YG and YO crystals and confirm their single-crystal-to-single-crystal (SCSC) interconversion.

While the color changes of single-crystal specimens can be precisely recognized in spectroscopy due to a certain wavelength separation by ca. 45 nm between the tops of sharp emission peaks from YG and YO crystals, the eye's visual perception of these changes is sometimes affected by the manner of lighting on the crystal specimens.

As mentioned in A.1-2, superelasticity in a YG crystal showing a clear emission color change demonstrates that anisotropic mechanical stress can mediate the crystal phase transition in accompanying large molecular movements, which is difficult to achieve thermally. Thus, superelastochromism is applicable to "molecular level" anisotropic displacement (or strain) sensing and can be regarded as a novel mechanism for developing color switching materials with high susceptibility to mechanical stress and spontaneous shape and color recoverability via SCSC phase transition. Note that there have been reports about stimulus-induced large emission color changes of molecular crystals via spontaneously irreversible SCSC phase transition (e.g., Ito, H. et al., *Nat. Commun.* 4, 2009 (2013)), suggesting a potential large color change by superelastochromism (spontaneously reversible SCSC phase transition).

Q.1-4. 4) Critically, all the experimental sections are way too brief missing important details that prevent other researchers in the field from reproducing their results. The manuscript is not suitable for publication until the methods are described with sufficient detail.

A.1-4. We have inserted additional information into the experimental section of the supplementary information to help readers comprehend the experiments:

1. Details on the preparation of **7Cl** crystals in YG and YO forms for single-crystal X-ray diffraction measurements, thermal analysis, and shear tests
2. DSC (Table S1) and force (Table S2, Fig. S1) measurement conditions along with figures showing the experimental setup

Q.1-5. 5) The authors claim exceptionally high effective shear stress and energy storage density compared with previously reported superelastic crystals. These data are based on a single measurement on one single crystal. Statistical analysis of these data as well as all other metrics in the same paragraph must be presented.

A.1-5. In fact, we did confirm the reliability and reproducibility of the values in our force measurements through a comparison with two more YG crystals (YG-2, YG-3) with different dimensions (please see Tables A and B and Figure A below). Mechanical properties were comparable to each other based on the estimated shear stress, which is a material constant.

Table A. Experimental information from shear tests on 7Cl crystals.

Specimen	Conditions		Crystal dimensions	
	Temperature / °C	Light	Width / μm	Thickness / μm
YG-1 (in this paper)	20	PW	246	20
YG-2	20	PW	258	33
YG-3	20	PW	406	25

Figure A. Stress-displacement curves of superelastic deformation of 7Cl crystals in YG form.

Table B. Mechanical parameters of YG crystals under various conditions.

Specimen	YG-1	YG-2	YG-3
Light	PW	PW	PW
Temperature (°C)	20	20	20
σ_f (MPa) ^a	1.527	1.464	1.456
σ_r (MPa) ^b	0.657	0.723	0.635
σ_c (MPa) ^c	1.092	1.094	1.046
E_s (kJ m ⁻³) ^d	553.6	610.9	535.0
E_d (kJ m ⁻³) ^e	733.0	622.2	692.1
η^f	0.430	0.495	0.436
χ^g	0.505	0.556	0.510

^{a,b}Effective shear stress for forward (σ_f) and reverse (σ_r) superelastic deformation. ^cChemical shear: $\sigma_c=(\sigma_f+\sigma_r)/2$. ^{d-}

^eEnergy storage density: $E_s=W_{out}/V$, dissipated energy density: $E_d=W_{in}/V$, energy storage efficiency: $\eta=W_{out}/W_{in}$, and

^gsuperelastic index: $\chi=E_s/\sigma_c$, respectively, where W_{out} , W_{in} , and V represent output work, input work, and volume of a deformed region, respectively, during superelastic deformation of YG crystals.

Q.1-6. 6) Since the superelasticity in monotropic system is new, it would be better to show schematic Gibbs free energy vs. variants (e.g., T) diagram in comparison to the enantiotropic system.

A.1-6. We have added a schematic diagram (Fig. S10) of Gibbs free energy vs. variants (e.g., T) in the monotropic system of **7Cl** crystals to the supplementary information, comparing that system to the enantiotropic system, e.g., crystals of terephthalamide,^{S1} tetrabutyl-*n*-phosphonium tetraphenylborate,^{S2} and aliphatic acids.^{S3} The Gibbs energy gap between YG and YO crystals (Fig. S10a) can be depicted as almost constant against temperature changes because no thermally-induced phase transition besides the shear-induced phase transition took place in our practical temperature range (at 0°C, 20°C, and 100°C) during the mechanical (superelastic) measurement.

Figure S10. Schematic representation of estimated Gibbs energy. (a) Gibbs energy diagram of a YG crystal (G_{YG}), YO crystal (G_{YO}), and liquid phase (G_{liq}). (b) Gibbs energy diagram of a crystal showing superelasticity by mechanically-induced phase transition (high-temperature phase: G_H , low-temperature phase: G_L , liquid phase: G_{liq}).

S1. Takamizawa, S. & Miyamoto, Y. Superelastic organic crystals. *Angew. Chem. Int. Ed.* **53**, 6970–6973 (2014).

S2. Takamizawa, S. & Takasaki Y. Shape-memory effect in an organosuperelastic crystal. *Chemical Science* **7**, 1527–1534 (2016).

S3. Takamizawa, S. & Takasaki Y. Versatile shape recoverability of odd-numbered saturated long-chain fatty acid crystals. *Cryst. Growth Des.* **19**(3), 1912–1920 (2019).

Q.1-7. 7) The authors provided crystal structure of before and after phase transition to show rearrangement of molecules in these structures. As the authors described, molecular packing change attributes emission property alteration, which is indicative of displacive motion of molecules in addition to 68° rotation of molecules upon transformation. The displacive term is not clearly stated in the manuscript.

A.1-7. We clearly described the molecular movements including displacive motion in the manuscript (please refer to A.1-2). In this paper, these events at the interface upon transition are discussed with merged pictures of YG and YO crystals based on single-crystal X-ray diffraction analyses under the superelastically deformed state.

Reviewer #2 (Remarks to the Author):

Thank you for your helpful comments for improving our paper.

Q.2-1. In this paper authors report two polymorphic forms of ESIPT active 7-chloro-2-(2'-hydroxyphenyl)imidazo[1,2-a]pyridine and elastic behavior of one polymorphic form. Authors claim that YG form get converts to YO form and vice versa. While YG form shows elastic nature, the other form does not. Authors claim that upon elastic stretching the YG form trnasforms to YO form (which I am not convinced with). Also, Mechanochromism in crystals via elastic deformation has already been known in the literature (authors cited some of the examples).

Different emission of different polymorphs of ESIPT active 2-Hydroxy imidazopyidinyl molecules are already reported by these authors (Angew. Chem. Int. Ed. 2008, 47, 9522). This polymorph dependent luminescence switching was done by heat as the stimuli. In the current paper mechanical force has been used as the trigger for polymorphic interconversion, which is a very well-known trigger for polymorphic transitions. This paper does not have the quality and scientific vigor required for a Nature communication article.

A.2-1. (From what you said about elastic deformation, we perceive your words “elastic stretching” to mean “superelastic deformation” by mechanically-induced phase transition from YG to YO.)

Superelasticity is a kind of diffusion-less plastic deformation, and it is completely different from elasticity (energy elasticity and entropy elasticity) in physics. In superelastic deformation, a crystal-to-crystal transition (at least locally) involving phase transition or twinning accompanying an orientation change compensates for the received mechanical strain. In ideal cases with single-crystal specimens, beautiful crystal bending can be observed through the generation of planar interfaces between the polymorphic phases or twins, and the structural change can be determined by (conventional) single-crystal X-ray diffraction techniques. Note that the superelastic bending, and thus phase transition or twinning, is spontaneously reversible whereas other known mechanochromisms (which, of course, are not superelastic) based on phase transition need other kinds of stimuli for recovery.

We have inserted phrases and amended sentences to touch upon the above thoughts.

(Please refer to A.1-2 for an explanation of superelastic transition between two monotropic polymorphs.)

Q.2-2. The elasticity in crystals is well known. I am surprised to see authors have not cited the work of Desiraju group and Malla Reddy group who have published several papers on elastic crystals.

A.2-2. Although superelasticity is a completely different physical property from the more commonly known elastic deformation of solids including organic crystals, we have inserted sentences briefly touching upon the research on elastic organic crystals. In addition, multiple related references were cited.

“Elasticity is a common physical property in the spontaneous shape recoverability of materials. Recently, research on the manner of elastic deformation of organic crystals has been intensive [16,17,18,19,20] despite a general perception of their brittleness. In contrast, superelasticity or more specifically plastic deformation with spontaneous shape recoverability is a minor and unusual physical property except in special kinds of metallic solids called “superelastic alloys” and “shape-memory alloys” [21,22] and research is still in its infancy especially in organic crystals. [15,23,24,25,26,27,28,29,30] In the elastic deformation, strain accumulates from density gradient variations of components, whereas the superelastic deformation can accommodate strain through orientation changes of domains upon phase transition or twinning; thus, superelastic deformation has a potential ability to abruptly change physical properties.”

[16] Ghosh, S. & Reddy, C. M. Elastic and bendable caffeine cocrystals: implications for the design of flexible organic materials. *Angew. Chem. Int. Ed.* **51**, 10319–10323 (2012).

[17] Worthy, A., Grosjean, A., Pfunder, M. C., Xu, Y., Yan, C., Edwards, G., Clegg, J. K., McMurtrie, J. C. Atomic resolution of structural changes in elastic crystals of copper(II) acetylacetonate. *Nat. Chem.* **10**, 65–69 (2018).

[18] Dey, S., Das, S., Bhunia, S., Chowdhury, R., Mondal, A., Bhattacharya, B., Devarapalli, R., Yasuda, N., Moriwaki, T., Mandal, K., Mukherjee, G. D. & Reddy, C. M. Mechanically interlocked architecture aids an ultra-stiff and ultra-hard elastically bendable cocrystal. *Nat. Commun.* **10**, 3711 (2019).

[19] Saha, S., Mishra, M. K., Reddy, C. M. & Desiraju, G. R. From molecules to interactions to crystal engineering: mechanical properties of organic solids. *Acc. Chem. Res.* **51**, 2957–2967 (2018).

[20] Varughese, S., Kiran, M. S. R. N., Ramamurty, U., Desiraju, G. R. Nanoindentation in crystal engineering: quantifying mechanical properties of molecular crystals. *Angew. Chem., Int. Ed.* **52**, 2701–2712 (2013).

[21] Otsuka, K. & Ren, X. Recent developments in the research of shape memory alloys. *Intermetallics* **7**, 511–528 (1999).

[22] Otsuka, K. & Ren, X. Physical metallurgy of Ti–Ni-based shape memory alloys. *Prog.*

- Mater. Sci.* **50**, 511–678 (2005).
- [23] Takamizawa, S. & Takasaki, Y. Superelastic shape recovery of mechanically twinned 3,5-difluorobenzoic acid crystals. *Angew. Chem. Int. Ed.* **54**, 4815–4817 (2015).
- [24] Takasaki, Y. & Takamizawa, S. Active porous transition towards spatiotemporal control of molecular flow in a crystal membrane. *Nat. Commun.* **6**, 8934 (2015).
- [25] Takamizawa, S. & Takasaki, Y. Shape-memory effect in an organosuperelastic crystal. *Chem. Sci.* **7**, 1527–1534 (2016).
- [26] Takamizawa, S., Takasaki, Y., Sasaki, T. & Ozaki, N. Superplasticity in an organic crystal. *Nat. Commun.* **9**, 3984 (2018).
- [27] Takamizawa, S. & Takasaki, Y. Versatile shape recoverability of odd-numbered Saturated Long-Chain Fatty Acid Crystals. *Cryst. Growth Des.* **19**, 1912–1920 (2019).
- [28] Sakamoto, S., Sasaki, T., Sato-Tomita, A. & Takamizawa, S. Shape memorization of an organosuperelastic crystal via superelasticity—ferroelasticity interconversion. *Angew. Chem. Int. Ed.* **58**, 13722–13726 (2019).
- [29] Sasaki, T., Sakamoto, S. & Takamizawa, S. Twinning organosuperelasticity of a fluorinated cyclophane single crystal. *Cryst. Growth Des.* **10**, 5491–5493 (2019).
- [30] Sasaki, T., Sakamoto, S., Takasaki, Y. & Takamizawa, S. A multidirectional superelastic organic crystal via versatile ferroelastic manipulation. *Angew. Chem. Int. Ed.* Accepted (DOI: 10.1002/anie.201914954).

Q.2-3. The abbreviation for room temperature is r.t. not RT (RT is the Product of Universal gas constant and absolute temperature)

A.2-3. Thank you for your kind indication. We have replaced “RT” with “r.t.” in the text of the manuscript according to your suggestion while leaving “RT” as is in Figure 4 (the term is defined in the figure’s caption) and videos for visual clarity. Having seen “RT” used as the abbreviation for room temperature in past issues of *Nature Communications*, we believe the use of “RT” for “r.t.” in the images does not violate the editorial/publishing policies of the Nature Publishing Group.

Q.2-4. Authors state that “Interestingly, the different fluorescence, yellow green and orange, exhibited by the α YG and β domains, respectively, under UV light suggests that the β domain is the YO (β YO) crystal (Fig. 3a, Mov. S2)”. But, as per the video S2 and Fig. 3a, beta domain does not exhibit orange fluorescence (more convincing evidences may be required).

How authors have determined the crystal structure of beta YG form? How authors got the elastically stretched form for crystal structure determination is not clear. The

details of the data collection will help other researchers to reproduce this results.

Also, even if it is showing orange fluorescence, how can authors say it is the YO crystal?

The figure 3 is just based on speculation. Are there any evidences ?

A.2-4. (Like in A.2-1, we perceive your words “elastically stretched” to mean “superelastically deformed” by mechanical shear force.)

The YG and YO crystal domains were characterized by fluorescence spectroscopy and single-crystal X-ray diffraction. Measurements on both a mechanically deformed YG crystal (coexisting α_{YG} and β_{YO} state) and YO crystal (coexisting α_{YO} and β_{YG} state) support the generation of YG from YO and YO from YG (Figures S4 and S5, Tables S3 and S4). Please watch the new video showing clear color changes from yellow-green to orange during superelastic deformation (the first half of Mov. S1) and refer to the modified Fig. 3a.

(Please refer to A.1-3 for a discussion of the superelastic transition.)

Experimental details have been updated to describe crystal preparation for the single-crystal X-ray diffraction measurements. (Please refer to A.1-4 for more information.)

Q.2-5. Readability: this paper has poor readability which talks at length about the calculated data. Experimental data are minimum.

A.2-5. In this revision, we have improved the paper by adding detailed information and new experimental data related to experimental procedures, thermal analysis, and mechanical properties to the manuscript according to the reviewers’ valuable and helpful comments for improving the academic quality of the paper. Since the native-English speaking editor and technical writer who worked with us on our original submission has continued to provide his assistance in proofreading and rewriting as necessary this version, we believe the manuscript describes our results clearly with readability.

Reviewer #3 (Remarks to the Author):

Thank you for your kind and valuable comments.

It is an exciting article by Takamizawa et al, which describes the superelastochromic nature in a biphasic organic crystalline material. The authors elegantly present the mechanical force induced phase transformation of a luminescent YG (yello-greenish) polymorph to the YO (orange) polymorph in a single-crystal-to-single-crystal fashion, allowing the clear detection of the colour change. The structures of the two phases (using SCXRD technique) and the superelastic behaviour of the samples are very well characterized. The origin of superelasticity is nicely correlated to the underlying crystal structure of the two forms. The crystals are so sensitive to the mechanical force that even movement of an ant can be detected by the fluorescence sensing, which I feel would find attractive practical applications. Hence, I feel the quality of the article is suitable for publication in Nature Communications.

Q.3-1. I do not find any draw backs in the article, but feel that more citations related to mechanochromic luminescence (ML) needs to be added from other groups. For instance, the authors discussed about the ML from systems with phase transformations, but did not mention about those involving stress-induced defects (no-phase transformations) such as those involving avobenzene based brittle and plastic polymorphs, etc. Although, the article already covers many pertinent references for the context of the current work, the broader perspective covering other mechanically responsive crystals would enhance the interest.

I find the article free of errors and easy to follow.

A.3-1. Mechanochromic luminescence by the introduction of defects or by a polymorphic or crystal-to-amorphous phase transition was described as follows to more clearly present the broader perspective.

– The sentence “Mechanically-induced defects can also induce mechanochromism.[4]” was inserted on page 2, and the sentence “Mechanochromism based on a polymorphic phase transition[10] is spontaneously irreversible...” was changed to “Mechanochromism based on a phase transition, e.g., polymorphic[14] or crystal-to-amorphous[2,3], is spontaneously irreversible...”.

– In addition, four references involving mechanoluminescence were cited.

[3] Ito, H., Saito, T., Oshima, N., Kitamura, N., Ishizaka, S., Hinatsu, Y., Wakeshima, M., Kato, M., Tsuge, K., Sawamura, M. Reversible Mechanochromic Luminescence of $[(C_6F_5Au)_2(\mu-1,4-Diisocyanobenzene)]$. *J. Am. Chem. Soc.* **130**, 10044–10045 (2008).

[4] Krishna, G. R., Kiran, M. S. R. N., Fraser, C. L., Ramamurty, U., Reddy, C. M. The

Relationship of Solid-State Plasticity to Mechanochromic Luminescence in Difluoroboron Avobenzene Polymorphs. *Adv. Funct. Mater.* **23**, 1422–1430 (2013).

[7] Sagara, Y., Yamane, S., Mitani, M., Weder, C., Kato, T. Mechanoresponsive Luminescent Molecular Assemblies: An Emerging Class of Materials. *Adv. Mater.* **28**, 1073–1095 (2016).

[14] Ito, H., Muromoto, M., Kurenuma, S., Ishizaka, S., Kitamura, N., Sato, H. & Seki, T. Mechanical stimulation and solid seeding trigger single-crystal-to-single-crystal molecular domino transformations. *Nat. Commun.* **4**, 2009 (2013).

REVIEWERS' COMMENTS:

Reviewer #1 (Remarks to the Author):

This referee has thoroughly read the revised manuscript, the response letter and the supporting information. While several of the technical issues raised has been addressed adequately, the two most critical issues remain (proving monotropic relationship and demonstrating application in mechanical sensing). Further, the addition of a closely related reference by Ito et al. to the revised manuscript raises a serious concern over the novelty of this work (ref 14, pasted below). Due to these outstanding issues, this referee cannot recommend publication of this work in Nature Communications. However, if the novelty and potential impact of this work is convincingly demonstrated and the remaining technical issues fully addressed, the work may still be suitable for a high impact multidisciplinary journal like Nature Communications. I detail my comments below. [14] Ito, H., Muromoto, M., Kurenuma, S., Ishizaka, S., Kitamura, N., Sato, H. & Seki, T. Mechanical stimulation and solid seeding trigger single-crystal-to-single-crystal molecular domino transformations. Nat. Commun. 4, 2009 (2013).

1) Most crucial concern is the novelty of this work compared to ref 14. Ito and colleagues have demonstrated very large mechanochromic effect during single-crystal-to-single-crystal transition of an Au-CN complex back in 2013. This referee understands that the current manuscript is different in that combining mechanochromic property with superelasticity is still new. However, why is it so compelling and impactful to also incorporate superelasticity with mechanochromic behavior? The authors only raise the possibility of reversibly sensing mechanical forces (which is in fact not demonstrated in this work as pressure sensing devices). However, this referee is not convinced that superelasticity is a big plus to this potential application. First of all, the mechanochromic shift is rather small in this current manuscript (per my previous comment #3) in comparison to what is demonstrated by Ito and colleagues. This is most likely limited by superelasticity which cannot endure drastic changes in the crystal lattice as in Ito's case. Because of the small color change, the mechanochromic property is highly sensitive to the lighting conditions per the authors' response. For instance, the new movie S1 has a different appearance than Figure 3a in terms of color change. This will severely limit the robustness of mechanical force sensing. To convincingly demonstrate the utility and novelty of combining superelasticity with mechanochromic behavior, the authors must demonstrate a reversible force sensor over many cycles that has superior performance and sensitivity than existing mechanochromic materials. Otherwise, there is no real advantage of combining superelasticity with mechanochromic behavior beyond scientific curiosity.

2) Regarding my previous comment #1, the new DSC curves at 6 degree/min still shows a bump between 30-60C which was consistent with the previous data shown in the original manuscript. This feature was ignored again in the revised manuscript. This could be a phase transition. As commented before, alternative approach to prove absence of phase transition must be performed to substantiate this unique phenomenon. DSC alone is inadequate. Further, DSC should be adequately analyzed such as by plotting the derivative of the curve to reveal any small changes previously ignored.

Reviewer #2 (Remarks to the Author):

This is a nice paper reporting the reversible chromism shown by crystals of an ESIPT-active fluorescent molecule, 7Cl. Two polymorphic forms of this compound show different emission and they interconvert reversibly in response to on and off-stage of the stimuli by virtue of their superelasticity. The ability of these crystals to sense even the movement of a small organism suggest that this material will be of practical use. The explanations and supporting data are sufficient enough to reproduce the results. The literature citation is apt. I recommend acceptance of this nice paper.

Reviewer #1 (Remarks to the Author):

This referee has thoroughly read the revised manuscript, the response letter and the supporting information. While several of the technical issues raised has been addressed adequately, the two most critical issues remain (proving monotropic relationship and demonstrating application in mechanical sensing). Further, the addition of a closely related reference by Ito et al. to the revised manuscript raises a serious concern over the novelty of this work (ref 14, pasted below). Due to these outstanding issues, this referee cannot recommend publication of this work in Nature Communications. However, if the novelty and potential impact of this work is convincingly demonstrated and the remaining technical issues fully addressed, the work may still be suitable for a high impact multidisciplinary journal like Nature Communications. I detail my comments below.

[14] Ito, H., Muromoto, M., Kurenuma, S., Ishizaka, S., Kitamura, N., Sato, H. & Seki, T. Mechanical stimulation and solid seeding trigger single-crystal-to-single-crystal molecular domino transformations. Nat. Commun. 4, 2009 (2013).

Thank you for your valuable comment for improving our paper.

1) Most crucial concern is the novelty of this work compared to ref 14. Ito and colleagues have demonstrated very large mechanochromic effect during single-crystal-to-single-crystal transition of an Au-CN complex back in 2013. This referee understands that the current manuscript is different in that combining mechanochromic property with superelasticity is still new. However, why is it so compelling and impactful to also incorporate superelasticity with mechanochromic behavior? The authors only raise the possibility of reversibly sensing mechanical forces (which is in fact not demonstrated in this work as pressure sensing devices). However, this referee is not convinced that superelasticity is a big plus to this potential application. First of all, the mechanochromic shift is rather small in this current manuscript (per my previous comment #3) in comparison to what is demonstrated by Ito and colleagues. This is most likely limited by superelasticity which cannot endure drastic changes in the crystal lattice as in Ito's case. Because of the small color change, the mechanochromic property is highly sensitive to the lighting conditions per the authors' response. For instance, the new movie S1 has a different appearance than Figure 3a in terms of color change. This will severely limit the robustness of mechanical force sensing. To convincingly demonstrate the utility and novelty of combining superelasticity with mechanochromic behavior, the authors must demonstrate a reversible force sensor over many cycles that has superior performance and sensitivity than existing mechanochromic materials. Otherwise, there is no real advantage of combining superelasticity with mechanochromic behavior beyond scientific curiosity.

In this paper, we reported the first success in coupling superelasticity and mechanochromism. Thus,

superelastochromic with large mechanochromic shifts will be possible by the further molecular design. In superelasticity, mechanochromism can be induced at any position(s) at any time by mechanically well-regulated structural transition. Intensity ratio between two discrete colors changes by exactly following the propagation of mechanically induced domain. These characteristics differentiate superelastochromism from conventional chromisms, e.g. superelastochromism can detect *in-situ* mechanical stress quantitatively: detection of the moment when mechanical stress is applied and removed and how much work is done on a crystal specimen. In this case, the color changes without any chemical reaction and the specimen consists of pure and simple organic molecules.

In the mechanochromism reported by Ito and colleagues, spontaneously irreversible mechanochromism is proceeded through the unique chemical reaction with a gold-gold dimerization originated from the exceptional organometallic characteristics in the components. On the other hand, the reported superelastochromism in this paper performs spontaneous reversible mechanochromism through a small packing structural change upon stress-induced SCSC phase transition, which can enable high sensitivity to small stress.

We have modified the conclusion section to describe the features of superelastochromism more clearly. (See no.2 in The modifications below.)

2) Regarding my previous comment #1, the new DSC curves at 6 degree/min still shows a bump between 30-60C which was consistent with the previous data shown in the original manuscript. This feature was ignored again in the revised manuscript. This could be a phase transition. As commented before, alternative approach to prove absence of phase transition must be performed to substantiate this unique phenomenon. DSC alone is inadequate. Further, DSC should be adequately analyzed such as by plotting the derivative of the curve to reveal any small changes previously ignored.

We are confident in saying monotropic transition of **7CI** since there is no thermal phase transition between YG and YO forms. What you mentioned as a bump is originated from not material's nature (phase transition) but the instrument by unsuitable cooling, i.e. air cooling around room temperature. In fact, there is no such bump in the DSC charts at 5°C/min where the specimen was cooled using liquid nitrogen rather than air. Absence of thermal phase transition between YG and YO forms is also supported by constant wavenumber of the emission peak tops of YG and YO crystals despite temperature changes.

Reviewer #2 (Remarks to the Author):

This is a nice paper reporting the reversible chromism shown by crystals of an ESIPT-active fluorescent molecule, 7Cl. Two polymorphic forms of this compound show different emission and they interconvert reversibly in response to on and off-stage of the stimuli by virtue of their superelasticity. The ability of these crystals to sense even the movement of a small organism suggest that this material will be of practical use. The explanations and supporting data are sufficient enough to reproduce the results. The literature citation is apt. I recommend acceptance of this nice paper.

We appreciate the kind and positive comment.

The modifications

1. Add sentences for updating the reference to the ferroelasticity in luminescent crystals of organometallic compounds.

“Very recently, organoferroelasticity—diffusion-less plastic deformation leaving spontaneous strain without spontaneous shape recoverability—in luminescent organometallic crystals was reported.[32] The luminescence is stable and unchanged during the organoferroelastic mechanical twinning, giving no mechanochromism.”

[32] Seki, T., Feng, C., Kashiya, K., Sakamoto, S., Takasaki, Y., Sasaki, T., Takamizawa, S., Ito, H. Photoluminescent ferroelastic molecular crystals. *Angew. Chem. Int. Ed.* 2020 in press. (DOI: 10.1002/anie.201914610)

2. Modified the conclusion section to clarify the features in the concept of superelastochromism.

“Organosuperelasticity offers the advantage of reversible and strict crystal deformation in association with molecular rearrangement against applied force. Mechanochromism can be induced at any position(s) at any time by superelasticity, or mechanically well-regulated structural transition. These characteristics differentiate superelastochromism from conventional chromisms, e.g. superelastochromism can detect in-situ mechanical stress quantitatively: detection of the moment when mechanical stress is applied and removed and how much work is done on a crystal specimen. The advantage of”

3. Changed colors in figures avoiding use of green and red as much as possible to follow the guideline of Nature Publishing Group.

4. The subsection “Synthesis of 7-chloro-2-(2'-hydroxyphenyl)imidazo[1,2-a]pyridine (**7Cl**)” was added to the experimental section.

5. Figures of NMR spectra of **7Cl** were added to the supplementary information, Supplementary Figs. 18 and 19.